# Perspectives in the Development of Tools to Assess Vaccine Literacy

**DOI:** 10.3390/vaccines12040422

**Published:** 2024-04-16

**Authors:** Luigi Roberto Biasio, Patrizio Zanobini, Chiara Lorini, Guglielmo Bonaccorsi

**Affiliations:** 1Giovanni Lorenzini Foundation, Viale Piave 35, 20129 Milan, Italy; 2Department of Health Sciences, University of Florence, 50134 Florence, Italy; patrizio.zanobini@unifi.it (P.Z.); chiara.lorini@unifi.it (C.L.); guglielmo.bonaccorsi@unifi.it (G.B.)

**Keywords:** vaccine literacy, health literacy, vaccine hesitancy, attitudes, psychometric tools

## Abstract

Vaccine literacy (VL) is the ability to find, understand, and evaluate vaccination-related information to make appropriate decisions about immunization. The tools developed so far for its evaluation have produced consistent results. However, some dimensions may be underestimated due to the complexity of factors influencing VL. Moreover, the heterogeneity of methods used in studies employing these tools hinders a comprehensive understanding of its role even more. To overcome these limitations, a path has been sought to propose new instruments. This has necessitated updating earlier literature reviews on VL and related tools, exploring its relationship with vaccine hesitancy (VH), and examining associated variables like beliefs, attitudes, and behaviors towards immunization. Based on the current literature, and supported by the re-analysis of a dataset from an earlier study, we propose a theoretical framework to serve as the foundation for creating future assessment tools. These instruments should not only evaluate the psychological factors underlying the motivational aspect of VL, but also encompass knowledge and competencies. The positioning of VL in the framework at the intersection between sociodemographic antecedents and attitudes, leading to behaviors and outcomes, explains why and how VL can directly or indirectly influence vaccination decisions by countering VH and operating at personal, as well as at organizational and community levels.

## 1. Introduction

Vaccine literacy (VL) is the ability to find, understand, and judge vaccination-related information to make proper decisions about immunization [1]. According to Ratzan [2], VL “is not simply knowledge about vaccines, but also developing a system with decreased complexity to communicate and offer vaccines”. There are distinct levels of VL. One is the personal level, which pertains to individual skills. Another is the organizational level, which includes the different degrees of complexity within an organization focused on communication and vaccine practice. Along with these two levels, there is a third larger level, called population or VL community. Costantini et al. [3] state that “VL is contingent on personal circumstances as well as the broader societal context…”, while Soeprobowati et al. [4] report that “VL is a balance between individual, community, and population skills in the complexity system”. Other definitions include Badua et al.’s [5]: VL represents a “process of providing vaccine information, building communication, and increasing people’s engagement about vaccines”. Zhang et al. [6] defines VL as “an important ‘endogenous driver’ of people’s vaccine choices, overcoming vaccine hesitancy and increasing vaccination rates”.

A recent scoping review has been published to collect, analyze, and summarize available definitions of VL, and to propose a comprehensive one [7,8]. In summary, it has been proposed that VL is the degree to which people have the ability to obtain and understand information regarding vaccination and related services. It entails ‘knowledge’, ‘motivation’, and ‘competencies’ to access, understand, and critically appraise and apply information about immunization, vaccines, and vaccination programs, at personal, organizational, and community levels.

Vaccinology is not only the science of vaccine development; it is a very specific branch of medicine that deals with vaccines and immunization practices, including several biological and social sciences [9]. In fact, VL is linked to health literacy (HL), but the two realms only partially overlap. Competencies and knowledge about vaccines and vaccination are unique: even individuals with higher levels of HL may lack the necessary skills regarding vaccination. Furthermore, VL is connected more than HL to a phenomenon strictly related to vaccination, which is vaccine hesitancy (VH).

VH is defined as an attitude of postponing or refusing vaccines despite their availability. Different VH models have been proposed. To the initial ‘3Cs’ model [10] (including individuals’ confidence, complacency and convenience toward vaccination), additional factors have been added to provide a better explanation to VH in the context of a complex social system showing evolving concerns towards vaccines [11]. Other attitudes from personal and psychological perspectives have been included in expanded models, like the ‘5Cs’ (comprising calculation and collective responsibility, in addition to confidence, complacency, and convenience) [12], and the ‘7Cs’ adding two other factors (compliance and conspiracy) [13]. VL encompasses these elements in its ‘motivation’-related dimension, but it also includes other dimensions, like knowledge and competencies, that are not part of the psychological determinants.

A meta-analysis [14], two systematic [6,15], and a scoping review [16] have been conducted recently on VL, including related assessing tools, replying to questions about the VL levels in the population, its determinants, and its outcomes, before and amidst the COVID-19 pandemic. As reported in these reviews, various instruments have been developed to assess VL, although they are limited by the number and complexity of influencing factors of such a complex construct [6]. Therefore, there is a need to develop new tools for a more extensive assessment of VL.

The objective of this study is to suggest a framework including the different visible or latent factors underlying VL, as well as related variables, to be used as a basis of the development and validation of future assessment tools.

In pursuing this objective, an updated overview of existing research on VL and its measures has been performed, in addition to a post-hoc analysis of an earlier dataset.

## 2. Materials and Methods

To achieve the objective of the study, we have divided the methodology into three main steps:updating our previous scoping review [16].performing a post-hoc analysis of data from a survey conducted in mid-2020 [17] through mediation and factor analysis. In this process, variables were relabeled to enhance the understanding of their interrelationships.developing a theoretical framework based on the existing literature, a backward citation search, and the post-hoc analysis. This is followed by the proposal of a process for the creation and validation of new VL tools.

### 2.1. Step 1: Review Update

A review was performed to update our recent research [16], conducted according to the Preferred Reporting Items for Systematic Reviews and Meta-Analyses (PRISMA) guidelines [18,19]. Findings from that scoping review were supplemented by a new search, using the same strategy and databases (PubMed/MEDLINE, Embase, Web of Science, CINAHL, Scopus, and PsycINFO) collecting publications from 1 December 2022 to 1 December 2023, as previous searches have been conducted from inception to 1 December 2022. The following search string was used in PubMed: “vaccine literacy” OR “vaccination literacy” OR “vaccination health literacy” OR “vaccine health literacy”. For Embase, CINAHL, Scopus, and PsycINFO the search string was: “vaccin* literacy” OR “vaccin* health literacy”. To be included in this review, studies should have described a tool/instrument/questionnaire/measure explicitly assessing VL, reported a VL score, and reported at least one determinant or outcome of VL.

For determinants, we considered any sociodemographic variables that could influence VL. For outcomes, we considered any variable that can be influenced by VL, particularly beliefs, attitudes, behaviors, and knowledge of participants; these variables could contribute to specific outcomes or effects, such as vaccine hesitancy (VH) or acceptance, as well as vaccine uptake. When referring to mediators, we specifically denote any variable explicitly declared as such by the author; these variables influence the relationship between VL and their respective outcomes.

A data charting form with the following elements was drafted: Author, reference, and year of publication; title, country of the study, and kind of population enrolled; number of subjects enrolled, gender, and age; design and time period of the study; VL determinants (moderators); VL scales used; mediators (when reported); outcomes (dependent variables); and main findings. Data extraction was performed by two independent reviewers, then the results were compared.

For the updated review, statistical analysis was aimed at descriptively comparing the populations being studied and results, such as demographics and VL scores, with those reported in the studies previously published.

### 2.2. Step 2: Post-Hoc Analyses

In addition, we carried out a post-hoc analysis using the dataset of our survey conducted in 2020. This survey recruited 885 individuals from the general population, who filled out an online questionnaire to evaluate VL levels regarding COVID-19 (COVID-19-VLS) [17]. The data series from this study was chosen because the same assessment scale has been translated into different languages and used in various populations [16]. In the context of future tool development, we investigated the mediating role of VL between demographic antecedents and beliefs regarding general vaccination. Additionally, we conducted a more comprehensive factor analysis, which had not been previously undertaken, at a 5% confidence level. Principal component analysis was applied to determine a minimal number of items explaining a high amount of variability, followed by confirmatory factor analysis to confirm the adequacy of potential new scales. SPSS v27 software [20] was employed, together with the open source software Jamovi v2.4.11 to complement analyses with additional tests like the mediation model using the jAMM module [21]. This package allows estimation of the direct and indirect effects of independent variables on the dependent variables by also examining all paths of the mediation model components. The use of more software also allowed us to verify the consistency between findings.

To perform the post-hoc analysis, beliefs regarding the two statements included in the questionnaire (Appendix B) about the safety of vaccines (*‘I am not favorable to vaccines because they are unsafe’*) and need to be vaccinated (*‘There is no need to vaccinate as natural immunity exists’*) has been taken as measures of ‘confidence’ and ‘complacency’, respectively, and evaluated through a four-item scale. The answer to the question *‘Do you want to pay a fee to be vaccinated?’* has been considered as a measure of ‘convenience’ (evaluated through a nominal scale: possible replies: yes or no). For the purposes of the post-hoc analysis, the last seasonal flu vaccine received (self-reported) was considered as behavior/outcome.

### 2.3. Step 3: Theoretical Framework and Tool Development Path

Following the literature update and the post-hoc analysis, we created a theoretical framework based on the Health Literacy Skills Framework, by Squiers’et al [22] and the Paasche-Orlow and Wolf model [23]. Moderators, including both proximal and distal determinants, alongside potential mediators—i.e., variables explaining motives and mechanisms behind outcomes—were considered. This consideration was based on the literature search and according to psychological models, like the Health Belief Model [24] and the Protection Motivation Theory [25].

Then, a process to explain how a new tool could be developed and validated has been proposed, according to our point of view and experiences.

## 3. Results

### 3.1. Step 1: Review Update

In addition to those included in the earlier review [16], we found a total of 367 papers on PubMed and other databases, published from 1 December 2022 until 30 November 2023, out of which we included those where VL was assessed using specific tools. Of the 246 papers screened by title and abstract, 17 publications were included, as shown in Figure 1 (PRISMA diagram) and charted in Table 1. Other publications that focused on the use of HL assessment instruments were analyzed and commented on, but were not included in the review as they did not specifically address VL.

The 17 selected papers included diverse populations, as detailed in Table 1. Samples’ size ranged from 133 to 12,586 individuals. Participants were distributed across different classes of age, but they were mostly young female adults. Nine publications out of the 17 confirmed the association between VL and attitudes, vaccine acceptance, or intention to be vaccinated, while three showed an indirect, mediated effect. The majority of the papers focused on COVID-19, while others addressed flu and HPV vaccinations. One paper discussed digital VL, and another explored the association of VL with VH (Table 1).

The scales employed in the studies were mainly modified versions of those used to measure HL chronic patients [43], based on the three-level HL model proposed by Nutbeam [44], encompassing functional, interactive (or communicative), and critical levels. The HLVa-IT tool (Vaccine HL for adults in Italian) [45], later translated into English (HLVa), aims at measuring VL levels associated with routine vaccination in adulthood. It consists of five questions assessing functional VL (FUVL), in addition to five and four items for interactive and critical skills, respectively. The functional sub-scale engages the semantic system, while the interactive and critical subscales regard more advanced cognitive efforts. The interactive-critical part of literacy may also include abilities about eHealth and AI-based approaches. A similar construct measure has also been used in parents of children [42,46]. In all these tools, answers are rated on a forced four-point Likert scale, with a mean (±SD) score calculated (range 1 to 4) and treated as continuous. A higher value indicates a higher VL level.

Based on the same construct, a measure was also developed and largely used to assess specifically COVID-19 VL (COVID-19-VLS). In this tool, the interactive and critical subscales have been merged and identified as interactive-critical VL (ICVL), and the total number of questions were reduced from 14 to 12, to lessen redundancy. COVID-19-VLS also includes items to assess other variables (mediating factors, like beliefs and attitudes) and behaviors, like vaccine COVID-19 vaccine intention, flu vaccine acceptance, and self-reported uptake [16,17]. Two versions of this tool were used, before and after the SARS-CoV-2 vaccines authorization and availability. Both measures are reported in Appendix B.

The validation process of the above assessment scales in different languages is described elsewhere [16]. A study of the tool translated and adapted into Chinese has been published recently [39].

During the pandemic, other tools were developed that utilized the HLVa construct. However, some of these instruments had a reduced number of items or employed different scoring methods than the original instructions. As a result, making descriptive comparisons became challenging.

#### Findings from the Review Update

Regarding the correlation between VL and vaccine acceptance, in two web surveys [34,35], Maneesriwongul et al. have explored Thai parental attitudes and VL about COVID-19 vaccination. While nearly all parents of children under five years of age received their own vaccine, only 45% intended on vaccinating their child. Factors influencing vaccine intention included parental age, attitudes, advice from healthcare professionals, VL, and belief in vaccine effectiveness. In the other study, out of 542 parents of children aged 5–11, 59% intended to vaccinate their child. In both studies, influencing factors included child age, parents’ education, VL, and positive beliefs on the vaccine. The parents’ VL interactive/critical literacy skills were among the most significant factors influencing parents’ intention to vaccinate their children. As for the VL score observed in these two studies, FUVL was between 2.67 ± 0.69 and 2.8 ± 0.71, and ICVL was between 3.31 ± 0.51 and 3.3 ± 0.56 (score range 1–4). These values are consistent with those reported by the same authors in the validation study of the VL tool conducted on the Thai general population [47].

On the contrary, other studies like Iskender’s et al. [31] and Bektas et al. [28], conducted on Turkish university students and parents, respectively, did not confirm a positive association between VL’s ability and attitudes to get vaccinated against COVID-19, although in both studies, it was not clear which variable was considered the antecedent and which was the mediator (Table 1).

An average VL score has been calculated from seven out of the 17 included studies. The FUVL score was 2.50 ± 0.33, whereas ICVL was 3.03 ± 0.26. These values have a similarity to those that were reported in our previous scoping review (2.83± 0.25 and 2.92± 0.42, respectively) [16], despite lower functional levels, which is difficult to interpret due to the inhomogeneity of the studies. Moreover, the VL scores reported in the other studies are not comparable, as they are not calculated as per the original instrument’s instructions [45]: in some studies, the score range considered was 1 to 5, rather than 1 to 4, or a summative score was used instead of the mean score (Table 1).

The role of the different variables as mediators between VL and outcomes was evaluated in three papers using VL tools [8,33,37]. Shon et al. [37] investigated the relationship between flu VL, health beliefs, and influenza vaccination. VL was assessed through the concept of understanding information, by administering one single question: “*Based on given information, do you feel sure about the best choice (vaccinated versus non-vaccinated) for you to prevent flu infection?”* (response options were yes or no). It was shown that flu VL affected both flu vaccine uptake and health beliefs assessed by the Health Belief Model. Analysis confirmed a mediating effect of health beliefs (perceived benefits, severity, and susceptibility) between VL and vaccination.

Again, with regard to the mediating effects, Collini al [8] showed that confidence in the flu vaccine (measured through the Vaccine Confidence Index—VCI) completely mediated the relationship between ICVL (assessed through HLVa) and the intention to get vaccinated, with significant effects observed in different population subgroups.

Similarly, Lu et al. [33] have described how the “3Cs” psychological antecedents of vaccinations (confidence, complacency, convenience) can mediate the relationship between VL (mainly ICVL) and VH assessed through a specific 10-items scale. Results also confirmed that higher VL is associated with lower hesitancy, and time-to-event analysis showed that participants with increased VH had a longer delay in vaccination.

The results of this literature update, along with those of previous reviews, have been taken into account to construct the theoretical framework.

### 3.2. Step 2: Post-Hoc Analysis

#### 3.2.1. Mediation

The post-hoc analysis of our 2020 survey data (N = 885) [17] was conducted using a multi-mediation model [21], including the educational level, sex, being a healthcare provider, and class of age as covariates, together with FUVL and ICVL. The results showed a significant mediating effect of ICVL between the independent variable ‘education’ and each of the ‘3Cs’ (accounting for 32%, 25%, and 57% of the total effect for confidence, complacency, and convenience, respectively, *p* between = 0.002 and <0.001, bootstrapped C.I. 95%, 1000 samples). The only significant indirect effect of FUVL was between education and confidence, accounting for only for 11% of the total effect (*p* = 0.032).

For their part, all the “3Cs” components proved to be significant, although partial mediators of ICVL toward the outcome ‘seasonal flu vaccine uptake’ (confidence 37%, complacency 26%, and convenience 33% of the total effect, *p* between = 0.002 and <0.001), while the only significant mediator of FUVL was confidence (14%, *p* = 0.025).

The introduction of the different covariates in the model did not substantially change the results, except for a moderator effect of increasing age on the relationship between FUVL and confidence toward influenza vaccine uptake (*p* = 0.014). This is in agreement with the idea that older individuals may develop more confidence than young people because of the increased experience with vaccination practice and recommendations [48].

Data are included in the Appendix A.

#### 3.2.2. Factor Analysis

From the same mid-2020 survey dataset [17], we have entered the 12 VL items in Principal Component Analysis, along with other variables placed between determinants and outcomes of the framework (Figure 2), namely, two items about beliefs and five about attitudes, for a total of 19 variables. The first two factors of the model explained 38% of the total variability (initial Eigenvalues, based on values > 1), and 35% after Varimax rotation (see Appendix A). Iterative analyses were conducted to reduce the pool of 19 to six items, based on the screen plots, loadings, communalities, and potential relevance to the various underlying dimensions. Before reduction, the first factor, including five items, corresponded to the VH ‘3Cs’ in addition to the intention to be vaccinated against COVID-19; the second factor (eight items) corresponded to ICVL; and the third (four items) represented FUVL, while the remaining items were more dispersed.

After reduction to six items, the first two factors of the model explained 63% of the total variability both before and after Varimax rotation, still maintaining an acceptable internal consistency of the dataset, assessed using McDonald’s ω. Confirmatory factor analysis showed an acceptable data–model fit [49], with the commonly used measures, CFI, SRMR, and RMSEA values going from 0.919, 0.0525, and 0.0589, respectively (19 items), to 0.984, 0.0221, and 0.0653 (six items). The combination of these techniques has already been used in the field of vaccination [50,51]. We found it important to check if confirmatory factor analysis could substantiate the construct suggested by principal component analysis, although it is executed on the same population.

Data is contained within the Appendix A.

Like the literature update, the results of the post-hoc analysis, particularly the mediation data, were useful for building the framework, while those of the principal component analysis served in proposing the composition of new possible evaluation tools, as described later.

### 3.3. Step 3: Theoretical Framework

Following the above steps, a logical framework was developed, which depicts the relationship between knowledge, motivation, and competencies incorporating them along with functional, interactive, and critical VL levels. Moderators (proximal and distal determinants) and possible mediators (i.e., variables that explain the reasons and mechanisms behind outcomes) were included as well. These variables include communication, knowledge, beliefs, attitudes, behaviors, self-efficacy, and competencies (Figure 2).

**Figure 2 vaccines-12-00422-f002:**
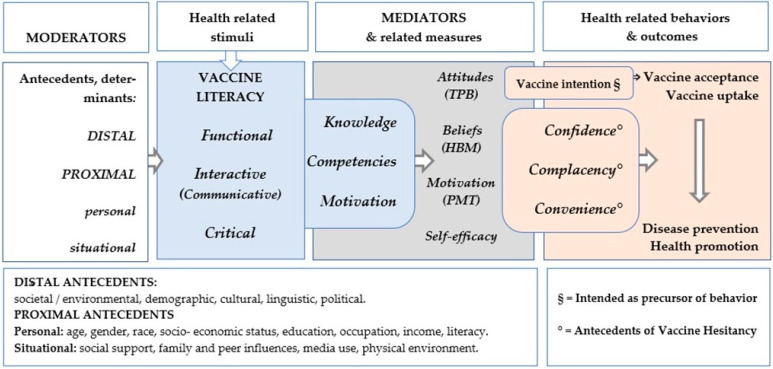
VL theoretical framework showing constructs, their influence, and measures. VL is placed between background (moderators) and mediators and partially overlaps these last, explaining its mediating and mediated role toward attitudes, behaviors and health outcomes. Adapted from Squiers’ Health Literacy Skills Framework [22] and Paasche-Orlow [23]. HBM = Health Belief Model [24,37,52], TPB = Theory of Planned Behavior [53], PMT = Protection Motivation Theory [54], Self-Efficacy Scale [55].

## 4. Discussion

### 4.1. The Role of Vaccine Literacy toward Vaccine Hesitancy, as Shown by the Updated Review

As mentioned, nine studies included in the updated review support the notion of a direct negative association between VL and VH [26,27,30,32,34,35,38,39,40], while the other three showed a partial or complete mediating effect [8,33,37]. These findings were contradicted by other investigations published during the same period [28,31]. Notably, these proportions are similar to those of our previous review [16], and of two systematic reviews, where 10 out of 13 papers reported a positive association between VL and COVID-19 vaccine acceptance [15], and 18 out of 21 showed an association between VL and acceptance, positive attitudes, or beliefs toward different vaccines [6].

Since published studies are cross-sectional online surveys with a one-time measurement of VL levels, it is difficult to infer precise causalities. Furthermore, their heterogeneity prevents comparisons in terms of methods used, results, and variables that may have influenced interpretation. For all these reasons, understanding the impact of VL remains challenging, although current literature is substantially more in favor of a relationship between higher VL levels and vaccine intention, acceptance, or uptake. Nonetheless, these findings allow considerations on the association of VL with other variables, its mediating role, and prospects for future research.

At the time when this paper was drafted, a meta-analysis has been published on the relationship on VLs, and vaccine intention, acceptance, or uptake, conducted on 18 studies, most of which already reported in previous reviews [14]. The results confirmed that VL significantly predicts vaccination intention, although its correlation with vaccination status (vaccine uptake) was comparatively weaker, despite being evidenced in several publications [6,15,16]. Notably, these results fit the proposed framework, as they highlight the position of VL between antecedents (moderators) and mediating variables, suggesting its indirect, and/or direct effect on behaviors and outcomes.

### 4.2. The VL Mediating Role: Literature and Post-Hoc Analysis

The effects of HL and VL on outcomes when mediated by other variables have been investigated, although with different objectives and methods. Using an 11-item tool, including general questions about immunization, Jiang et al. [56] showed that the relationship between perceived HL and COVID-19 vaccine acceptance was completely mediated by attitudes toward general vaccination and self-efficacy of the COVID-19 vaccine. The pathway from determinants to vaccine acceptance has been shown by Hurstak et al. [57,58]. In a population of adults, using a functional HL tool (Touchscreen Technology-LiTT) and a vaccine confidence scale, the authors revealed that HL mediated the relationship between demographic variables and vaccine confidence, which in turn mediated the relationship between HL and COVID-19 vaccine acceptance.

As reported in the results, using a VL single item nominal tool, Shon et al. [37] demonstrated the mediating effects of health beliefs between flu VL and flu vaccine acceptance in students, although the literacy of influenza vaccines also directly improved the vaccination behavior. Collini et al. found that vaccine confidence completely mediated the relationship between ICVL (assessed through the HLVa tool) and the intention of nursing home personnel to get vaccinated against flu [8].

Finally, according to Lu et al. [33], using the COVID-19-VLS tool in the Chinese general population, all the psychological antecedents of the ‘3Cs’ model played a significant role in mediating VL with VH, accounting for 66% and 95% of the total effect of FUVL and ICVL, respectively. Using the same tool to check mediating effects on our previous data series (unpublished data) [17], we found that in the Italian general population, VL (in particular ICVL) significantly mediated the relationship between demographic variables such as education, and positive beliefs about vaccination, reflecting confidence, complacency and convenience. In their turn, the same beliefs partially, but significantly mediated the relationship between ICVL and flu vaccine uptake.

Comparing different mediation models is challenging, and it should be carried out carefully. However, the above observations show that VL can significantly affect the acceptance of the vaccine both directly and indirectly, likely more than HL. It was reported [59] that there was no significant correlation between flu vaccine uptake and HL when evaluated by a general functional tool (Imeter). These findings were confirmed in another survey using the same functional tool, where an association was found between vaccine intention and the vaccine confidence score, but not with the HL score [60]. Similarly, HL did not affect the likelihood of influenza vaccination uptake and did not mediate the relationship of any independent socio-demographic variable with flu vaccination in high-risk individuals, using the six-item European Health Literacy Survey Questionnaire, not including specific items on immunization [61,62].

In summary, available data suggest that specific VL assessments—especially ICVL—contributes to analyzing VH predictors. As mentioned, in addition to its conceptual definition, VL can be explained by its ‘placement’ in the framework, as it sits at the intersection between antecedents (moderators) and intermediate variables (mediators), partially overlapping these last (Figure 2). The more the VL tools entails elements belonging to the mediation area (beliefs, attitudes, motivations, self-abilities), the more VL shows a greater impact as a driver towards vaccine intention (considered as precursor of behaviors [63]) and towards health behaviors and outcomes, such as vaccine acceptance (i.e., the degree to which individuals accept or refuse vaccination [64]) and vaccine uptake (number of individuals actually vaccinated [65]). This is in the context of the specific domains of VL, disease prevention, and health promotion (Figure 2).

### 4.3. Proposal of New Tools, Based on the Theoretical Framework

#### 4.3.1. Current Tools

The VL instruments developed so far are intended for the assessment of the personal VL of the adult general population, although some have been adapted to select adult populations. As mentioned, these tools include items related to functional and interactive-critical VL.

Functional literacy refers to the use of semantic and cognitive abilities like reading, writing, knowledge of medical terms, and mathematics [66]. The quantification of these abilities is possible using performance-based tools. Inversely, self-reported measures typically evaluate the psychological aspects that underlie components like motivation, beliefs, attitudes, and the ability to engage with information and make decisions (listening, speaking, interpreting). Standardized questions are used in objective assessment, while subjective measurements involve, typically on Likert scales, people self-reporting to questions on their experiences about health, although it is challenging to establish a connection between a person’s response and their actual skill [67].

Performance-based tools appear more suitable for estimating individuals’ skills in the health care domain, while self-reported measures are better for assessing individuals’ attitudes and knowledge beyond reading and numeracy, such as understanding the value of vaccination. Considering the domains relevant to VL (disease prevention and health promotion [68]), the use of subjective VL tools, like the HLS19-VAC Instrument for measuring vaccination literacy [69], seem appropriate for that scope. The same is true for HLVa and derivative measures, as they entail items related to motivation and competencies. Their construct has been validated in the general population of different regions [16]. Principal component analysis, as well as exploratory and confirmatory factor analysis, were used to extract the latent factors defining their construct. Through parallel analysis, different authors have identified two separate components underlying the FUVL and ICVL items, explaining high and comparable percentages of the total variance and significantly similar factor loadings [16].

It has been reported that some latent factors might be underestimated in current VL tools [6]. However, similar to HL [70], VL is a latent construct by definition: reliable assessment scales can be constructed and validated, although the results may represent aspects of VL without claiming to encompass its entirety. Indeed, we believe that despite the limitations of the current VL tools and the predominantly cross-sectional nature of surveys conducted thus far in the context of the pandemic, the accumulated experience remains important. The COVID-19 pandemic has likely influenced public sentiment towards the prevention of viral diseases, leading to long-term impacts in the way the general population perceives communicable diseases. The pandemic experience will in any case affect VL about other vaccines, at least in the near future. Thus, although the experience of VL tools used mainly during the COVID-19 outbreak can be considered limited, it provides a relevant reference for future research [16].

#### 4.3.2. Future Tools

Vaccination is a primary prevention practice, mainly aimed at healthy people, which may also require an assumption of responsibility and decisions on behalf of others (such as parents with respect to their children) [7]. Because of this, predictors of vaccine acceptance (like educational levels, socio-economic status, comorbidities, etc. [71]) may differ from those of other health behaviors. Similarly, the skills needed to navigate, understand, evaluate, and apply information related to immunization are likely to differ from those needed for other health issues [7]. These aspects must be taken into account in the development of new tools.

As for HL [70], new methods for VL measurement should explicitly refer to the domains outlined in relevant conceptual frameworks. In addition, to address the limitations of current measures and align them as much as possible with the most recent definitions of VL [7], the construction of new instruments should contain and integrate items related to all the three components (motivation, knowledge, competencies), thus reducing the risk of underestimating latent factors. We attempted to determine an effective approach for incorporating them along with the three VL domains (functional, interactive, and critical) within the Health Literacy Skills Framework illustrated in Figure 2. The framework proposed by Squiers [22] was used for our analysis, together with the Paasche-Orlow and Wolf model [23]. We considered moderators (proximal and distal determinants) and possible mediators (i.e., variables that explain the reasons and mechanisms behind outcomes). These variables include communication, knowledge, beliefs, attitudes, behaviors, self-efficacy, and competencies, incorporating concepts from various psychological theories, such as the Health Belief Model [24] and the Protection Motivation Theory [25], among others. We referred to tools designed for assessing these variables, relevant in the vaccination field, also taking into account the mentioned VH models (Table 2). Based on this, we have endeavored to outline potential new VL tools.

##### Motivation

There are different definitions for motivation [72]; according to the American Psychological Association [73], motivation refers to “a person’s willingness to exert physical or mental effort in pursuit of a goal or outcome” (Appendix A). It is a dynamic concept, resulting from internal and external inputs that lead to decisions and behaviors [72]. Motivation encompasses attitudes, which includes beliefs, emotions, and evaluations. In clinical settings, there are various tools for measuring motivation [72]. Considering the consistency of the results reported so far from studies using HLVa and derived measures (like COVID-19-VLS), the VL scales entailed in these tools appear suitable for assessing motivation also for future research, in particular in the interactive–critical subscale. It is also possible to design VL tools to assess motivation by adapting items from the Protection Motivation Theory, which explains how people respond to fear-evoking or threatening messages, or from the Health Belief Model, which explains and predicts health behaviors by examining the attitudes and beliefs as previously demonstrated by others [74]. Importantly, the papers on behavioral change models cited in Table 2 stand as just examples of very extensive literature.

##### Knowledge

Knowledge is the state of being familiar with something or aware of its existence, usually resulting from experience or study (i.e., information learned) [73]. It is part of the conceptual VL definition and a key mediating component in the VL framework. Procedural knowledge, which involves understanding how to carry out specific tasks or actions, serves as the basis for ICVL skills and fluid cognitive abilities, whereas crystallized abilities, including generalized knowledge and vocabulary, are typically more associated with functional skills, particularly in the elderly population [66]. Understanding the relation of VL to basic cognitive abilities is important since research has shown that both general intellectual abilities and literacy are related to health [75]. While IQ tests can provide a broad assessment of cognitive abilities, adding psychological assessments to the tools designed for the general population can be challenging. Evaluating the level of crystallized knowledge through a vaccine quiz can offer a straightforward, relevant, and efficient performance-based method. This approach has been previously employed for validating the theoretical construct of HLVa-IT, where the quiz was administered together with the VL questionnaire [76]. In addition, vaccine quizzes can be structured to objectively evaluate the individual’s functional reading and understanding skills when administered through face-to-face interviews or by paper-and-pencil. Regarding online surveys, there is an inherent risk that individuals may look for aid or refer to external sources, which can potentially inflate their scores on assessments. However, measures can be taken to reduce this risk, such as underlining the anonymity of the survey, or by designing the questionnaire in a linear, one-directional flow where respondents can only move forward, and by including control or confirmation questions.

The content of the scales can be adapted over time according to the vaccines that will become available in the future, although addressing the general knowledge about vaccination and on the most common routine vaccines (D, T, Polio, Influenza) would reduce disparities. Items can be selected from vaccine scales available in the literature [77], as well as online resources provided by academic [78] and international institutions [79]. Common questions on the knowledge of vaccines and diseases should be identified and used for comparability purposes, while taking into account cultural and socio-economic differences between populations.

##### Competencies

Competencies can be viewed as a set of knowledge, skills, capabilities (abilities), and behaviors that contribute to the individual’s performance [80]. Given that knowledge on vaccines can be evaluated as described above, and that skills and abilities can be assessed through the existing VL scales (such as the interactive and critical sub-scales of HLVa), it is suggested to complete the assessment of competencies by incorporating standardized items that evaluate intention to be vaccinated (intended as precursor of behavior, together with the educational level of respondents. This suggestion is based on the general understanding that individuals with higher education levels are more likely to adopt healthy behaviors [81]. Education plays a crucial role in providing individuals with the knowledge and skills required to develop competencies in various domains such as health, although HL is not solely dependent on educational levels, and these competencies can also be developed through other means in addition to education. Selected items from self-efficacy scales [55] specifically developed and adapted to the vaccination field may be useful in completing the assessment of competencies. However, it is important to underline that competencies refer to the actual knowledge and skills possessed by an individual, while self-efficacy relates to an individual’s belief in their capability to use their competencies effectively [73].

#### 4.3.3. Composite Tools

In the literature, composite instruments are reported, combining scales to evaluate different variables related to vaccination [12,82]. Creating multidimensional composite VL tools is challenging because of the complexity of influencing factors and the many concepts involved. Elements related to psychological content, such as motivation, beliefs, and attitudes, are often included in a variety of other tools usually administered together with HL or VL questionnaires in conducting investigations. Incorporating some of these elements in a single VL tool would standardize responses, making comparisons easier. Our proposal involves selecting a few items for each VL subscale and incorporating other questions related to beliefs, attitudes, and behaviors, in addition to knowledge. The selection of these elements can be guided by meaningful grouping [83], reliability, and factor analyses, in addition to specific statistical methods to handle categorical variables, like the two-stage path analysis [84]. These methods can ensure the construct validity of composite instruments, despite the reduction in the number of items.

Indeed, COVID-19-VLS, frequently utilized in online surveys [6,14,15,16], can already be considered a composite VL measure. In fact, in addition to the VL subscales, it includes questions related to beliefs and attitudes underlying confidence, complacency, and convenience, as well as to coronavirus vaccine intention and behaviors such as flu vaccine uptake (Appendix B). Although for COVID-19-VLS, a reduction in VL items was made compared to HLVa, we were able to further reduce their dimensionality by applying principal component analysis as a method of data reduction, as described above. This exercise was also performed by considering the results of the mediation analysis, balancing the weight of the single VL items with the purpose to ensure the use of similar elements in future research. This involves including additional questions on knowledge and skills to new assessment scales, maintaining an acceptable total number of items in order to balance the length of the questionnaire with people’s willingness to participate in the surveys. However, the decision on how many items to use should consider the trade-off between maintaining significant factors, reliability, and data interpretation. This approach will also be useful in addressing a limitation of the current VL instruments, related to a possible underestimation of their specific dimensions [6]. In fact, the current VL instruments are derived from tools that were originally developed to assess HL in other areas of medicine (chronic patients) [43], although they have been validated in various languages and cultural settings [16].

In summary, using similar metrics on all elements studied, balancing them at the same time and considering their association with each other, a composite-possibly unit-weighted VL score can be sought for future assessments. In addition to separately measuring each scale included in a multidimensional framework—useful to observe the correlations and the mediating effects of the variables between them—adding a standardized combined index would allow a simplified representation and easier interpretation of results, as well as improving statistical power [85]. Moreover, including self-reported and performance-based elements in the same tool would facilitate a better assessment of the overall individual VL levels without the need for additional tests. In fact, it has been shown that the combination of the results obtained using performance-based measures of functional HL and self-assessed measures of general HL may result in an increase in sensitivity (i.e., the identification of people with low HL skills) and improve the understanding of the relationship between HL and its antecedents [86]. It is reasonable to assume that the same can be valid for VL.

**Table 2 vaccines-12-00422-t002:** Psychological theories explaining behaviors towards vaccination and related assessment measures, usually rated through odd-points scales. Different examples of items reported in the literature (for influenza, COVID-19, and HPV) are proposed for each component of the models, to be possibly used or adapted for building new VL tools to explore motivation and competencies; knowledge can be assessed by performance-based measures, administering standardized stimuli.

Psychological Frameworks/Models									
**Self-Efficacy Theory VacSE** **Włodarczyk** [55] ***explains how individual’s belief in their own abilities drives to successfully perform tasks***		**Health Belief Model HBM** Conner [24], Shon [37], Carpenter [52] *explains and predict health behaviors by examining the attitudes and beliefs*		**Theory of Planned Behavior TPB** Ajzen [53], Wolff [63], Catalano [87] *explains how intention to engage in a behavior is influenced by the attitude towards that behavior*		**Protection Motivation Theory PMT** Marikyan [25], Kowalski [54] *explains how people respond to fear-evoking or threatening messages*		**‘3Cs’ and “5Cs”** McDonald [10], Betsch [12] Lu [33] *explains respectively 3 and 5 key determinants that contribute to VH*	
* **Models’ items and vaccine related examples of statements** *									
**To what extent are you sure that you will vaccinate in the current season even if…**	*“… you have to pay in full or in part for the influenza vaccination”*	**Perceived severity**	*“I am afraid the flu will make me very sick”*	**Attitudes**	*I think getting all three doses of the HPV vaccine within 12 months is …”* *very bad–very good, extremely harmful–extremely beneficial, unnecessary–necessary.*	**Perceived severity**	*“The negative impact of COVID-19 is very severe”*	**Confidence** **(see also §)**	*“Generally, I trust the information released by the state on a COVID-19 vaccine”*
	*“… friends or the media tell you that this flu vaccine is harmful or unnecessary, or that it is does not give a 100% guarantee”*	**Perceived benefits**	*“Flu vaccinations are an effective protection against the flu”*	**Subjective norms**	*“Most people who are important to me think that I should get all three doses of the HPV vaccine in the next 12 months”*	**Perceived susceptibility/ vulnerability**	*“If I don’t get the COVID-19 vaccination, I am at risk of catching the COVID-19 virus”*	**Complacency**	*“I’m healthy and resistant to infection, so I don’t have to get the COVID-19 vaccine”*
	*“… you will need to find out where and how to get the flu vaccine”*	**Perceived barriers**	*“Flu vaccination has unpleasant side-effects”*	**Perceived behavioral control**	*“If I wanted to, I am sure I could get all three doses of the HPV vaccine in the next 12 months”*	**Maladaptive response rewards MMR**	*“If I do not get a COVID-19 vaccine, I will not have to spend time and money getting vaccinated”*	**Convenience**	*“I don’t like going to medical facilities, so I’m reluctant to get the COVID-19 vaccine”*
	*“… you will be overwhelmed by the excess of other things and responsibilities”*	**Perceived susceptibility**	*“I have an increased risk of falling ill with flu”*	**Behavioral intention**	*“I plan to get all three doses of the HPV vaccine in the next 12 months”*	**Outcome efficaciousness**	*“I’m sure that having a COVID-19 vaccine would be effective in reducing my personal risk of contracting the virus”*	**Calculation**	*“When I think about getting vaccinated, I weigh benefits and risks to make the best decision possible”*
	*.”… vaccination will have to be rescheduled, for example due to a cold”*					**Self-efficacy**	*“I’d be able to get a COVID-19 vaccine if I wanted to”*	**Collective responsibility**	*“When everyone is vaccinated, I don’t have to get vaccinated, too”*
	*“… it will be necessary to make further attempts to make an appointment” ”during the pandemic”*					**Response cost**	*“Being vaccinated against COVID-19 is painful”*		

§ = There are several scales to assess VH [12]; a frequently adopted measure is VCI [88,89]. The flu example [8,59] assessed on 4-point Likert scale-VCI = [(A1 + A2 + A3 + A4)/4]/[(B1 + B2 + B3 + B4)/4]; (A1) Flu is a serious illness, (A2) Flu vaccine is effective, (A3) HCWs must get vaccinated, (A4) By getting vaccinated I can protect people close to me; (B1) It is better to contract flu than to get the vaccination, (B2) Flu vaccines have serious side effects, (B3) Vaccine can cause the flu, (B4) Opposed to flu vaccine.

#### 4.3.4. Specific VL Measures and Selected Populations

Questions included in new VL tools should be adapted to the specific context for which the measure will be intended. Unlike HL where there is a huge proliferation of measures [67], the number of tools for VL is relatively limited. Therefore, as HL tools are developed for several specific contexts and populations outside of pandemic emergencies, a similar approach should be adopted for VL. In addition, it will be important to develop dedicated tools for evaluating VL in different medical areas, such as routine vaccination for children, for patients, and other specific categories (e.g., healthcare workers and travelers), as well as for unique psycho-physiological situations—for instance, pregnancy, even though existing VL tools have been used in some of them [42,90,91,92,93]. Moreover, in the future, vaccine applications will extend beyond the prevention of infectious diseases. For instance, mRNA and siRNA techniques hold potential in various healthcare areas, including oncology and diseases with genetic components [94]. Therefore, the development of future VL instruments is likely to extend beyond the specific area of communicable diseases.

Adolescents are another critical area for future research. The pandemic has had many negative effects on teenagers, especially in low- and middle-income countries, and vaccine coverage rates against SARS-CoV-2 in younger age groups were insufficient even in developed places [95]. The controversial nature of coronavirus vaccination has exacerbated the pressure on parents who make decisions about their sons and daughters’ immunization. Yet, in recommending vaccines, it is important to consider not only parents’ attitudes to increase uptake, but also adolescents’ awareness of the infective risks, their knowledge of self-consent rules, and the relevance of taking part in vaccination decisions [96]. These aspects should be worth exploring through the development of specific VL measures. Over 40 tools are available for HL assessment in adolescents aged 10 to 7 years [97], but none are for VL, so far.

As mentioned, the available VL scales have shown good consistency in results across different countries, which has been proven not only by comparing the average scores observed between populations, but also by comparing factor analyses data. However, the cutoff values have been set only arbitrarily so far. Regarding the tools HLVa and COVID-19-VLS, VL has been defined as limited when the score is ≤2.5, or when belonging to the lower tertile of the average values observed in a given population [16]. For example, in the mentioned 2020 survey, the lower tertile bound corresponded to a score of ≤2.50 for FUVL and ≤3.13 for ICVL, which allowed us to define low-literate people, as already defined by others [92,98]. Using the lower tertile approach would more rigorously evaluate literacy levels according to local settings. In such a case, turning the score to a standardized one (observed value–mean of the sample/SD) could allow for meaningful comparisons between populations.

Validation of future VL tools should ideally be performed internationally to get results at the same time in more than one country, such as has been for the HLS19-VAC Instrument for measuring vaccination literacy [69], aiming to define a universally applicable threshold value. However, referring to local average scores to identify limited VL will remain. Future tools should also be proposed in prospective cohort and longitudinal studies for a better understanding of the causal relationship between VL and VH, and the relevance of the mediating role of VL. To improve homogeneity and comparability of populations, new tools should be administered via the web in a standard manner, trying to reduce biases related to online surveys–like the social desirability bias—as much as possible, using multi-item scales and combining self-report measures with other data sources, such as behavioral observations, as well as underlining anonymity. As mentioned, adding objective measures, such as about vaccine knowledge, will also be helpful.

Finally, while a definition for organizational VL has been proposed [7] and is already mentioned in the literature [99], it is important to develop specific measures for it. Improving organizational VL is crucial for increasing vaccination rates, as healthcare organizations play a vital role in providing trustworthy and accessible information to the community. Specific instruments to evaluate organizational HL capacities have already been created and applied [100,101]. It should be performed the same way for organizational VL, which should be the subject of dedicated research.

### 4.4. Limitations

Despite the use of various databases and attempts to be as comprehensive as possible, the updated literature review may not have identified all relevant recent articles, as the overall search strategy may have been biased toward public health. Searches of other databases may have resulted in other relevant publications. Furthermore, the search was conducted using only English terms, which possibly could have led to missing some studies. Limitations and the heterogeneity of online cross-sectional studies in terms of methods used and reporting of results may have affected the interpretation of the data. Furthermore, due to the heterogeneity of the results reported in the included studies, the findings were only addressed descriptively.

Many of the published studies on VL used the same scales (HLVa or COVID-19-VLS), although translated into different languages and validated in various populations. This is a limitation, considering the wide variety in rating scales of questionnaires in other areas. However, it can also be considered a strength because it can facilitate comparisons, not only in terms of VL levels and scores, but also in the exploration of the mediating effects of the different variables. Through associating various studies, we believe it has been possible to obtain a fairly accurate understanding of the current utilization of tools and the assessment of VL skills. This understanding remains largely descriptive, showcasing diverse values across different regions and populations. These variations are likely linked to methodological and/or local differences.

Finally, it could be seen as a limitation to rely on a post-hoc analysis of previous data series instead of analyzing new studies in order to discover a way to develop new instruments. However, it is important to note that the dataset used was from the initial study where the COVID-19-VLs tool was first utilized [17]. Moreover, being familiar with this data, we have found it to be a valuable exercise that will assist in creating new assessment scales for future studies. This perspective solely represents the viewpoints of the authors and their expertise in VL domains. Comments and proposals from other research groups, hopefully numerous, will be welcome to broaden the discussion and progress on this important public health topic.

## 5. Conclusions

The existing literature shows that the relation between VL and VH is uneven, although the majority of publications support a negative association. Current self-rated assessment tools, such as HLVa and derived measures, seem sufficiently adequate to measure VL skills, despite some specific dimensions may be underestimated due to the complexity of influencing factors. To reduce these limitations, we propose a framework and a path for the development of new tools to evaluate VL, based on existing studies, and tested by conducting a post-hoc analysis of our previous dataset. The framework aligns with the latest VL definitions, aiming to assess both knowledge and competencies in addition to the psychological components related to motivation. Future research should be focused on developing measures, including self-rated and performance-based items, where possible. Such measures will ease further research about the direct and the mediating role of VL towards outcomes and its relationship with VH, as well as on unexplored aspects, such as the longitudinal evolution of VL in different populations and contexts, and the application of research in organizational literacy. Furthermore, the assessment of VL across various categories of populations and patients, in addition to healthcare workers, can also be the target of new tools. A better understanding of the causal relationship between VL and vaccination will provide a better basis for communication and health education campaigns.

## Figures and Tables

**Figure 1 vaccines-12-00422-f001:**
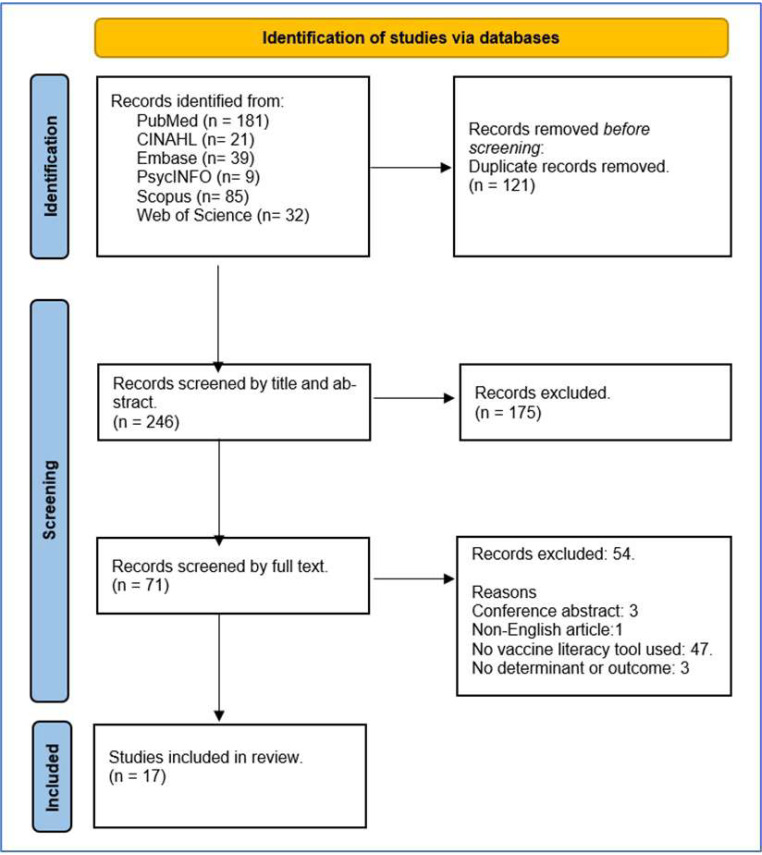
PRISMA diagram.

**Table 1 vaccines-12-00422-t001:** Included studies published in 2023 using VL tools.

Author, Ref #, Year	Title	Country, Study Population	Subjects N, Gender, Mean Age	Study Design, Period	Determinants in Addition to Age, Gender, Education	VL Tools, Score	Mediators	Dependant Variable(s)	Main Findings: Bold = Exploring Mediation *Italic =* *VL & VH Pos.Association*
Akova et al. [26] 2023	COVID-19 Vaccine Literacy and Vaccine Hesitancy Level Among Healthcare Professionals in Turkey, Their Relationship and Influencing Factors: A Cross-Sectional Study	Turkey, HCWs	1111, females 59.6%, mean age 34.3 ± 9.2 years	Online, cross-sectional, 15 February 2022–15 March 2023	Occupation, working time, area of residence, presence of chronic disease	COVID-19-VLS, FUVL 2.6 ± 0.7 ICVL 3.0 ± 0.6	VH	Opinions on coronavirus and COVID-19 vaccines	*High VL decreased VH*
Alyahya et al. [27] 2023	The Social Attitudes Towards the Booster Dose of the COVID-19 Vaccine and the Associated Factors Among Residents of Riyadh, Saudi Arabia	Saudi Arabia, residents 16+ years old	435, females 72.6%, mean age 38.1 ± 13.6 years	Online, cross-sectional, from 22 August 2022 to 25 August 2022	No association investigated	HLVa- functional 80.3% > 2 critical 77.4% > 2 communicative 78.3% > 2	None investigated	VH	*VL washigher* *in non hesitants, although* *not statistically significant*
Bektas et al. [28] 2023	The effects of Parents’ Vaccine Hesitancy and COVID-19 Vaccine Literacy on Attitudes toward Vaccinating their Children During the Pandemic	Turkey, parents of children aged 0–18	199 female 87.9% mean age 38.74 ± 6.39 years	Cross-sectional online	HCWs, income, number of children and age, children & parentsì Covid disease & vaccination status	COVID-19-VLS Score not reported	None investigated	Parents’ attitudes toward getting children vaccinated, VH Scale 10 items, 2 sub dimensions (Larson)	VH Scale alone significantly affected attitudes during the pandemic. VL did not affect the parents’ attitudes toward vaccinating children
Bellomo et al. [29] 2023	Who Chooses Alternative Sources of Information about Childhood Vaccinations? A Cross-Sectional Study	Italy, parents	2301, females 81%, mean age 47.7 ± 6.4 years	Online, cross-sectional, from June to October 2021	No association investigated	HLVa-IT, functional 80.3% > 2 critical 77.4% > 2 communicative 78.3% > 2	None investigated	Use of alternative information sources	Parents with lower HLVa score more inclined to use alternative sources of information
Collini et al. [8] 2023	Does Vaccine Confidence Mediate the Relationship between Vaccine Literacy and Influenza Vaccination? Exploring Determinants of Vaccination among Staff Members of Nursing Homes in Tuscany, Italy, during the COVID-19 Pandemic	Italy, nursing homes Staff	1794, females 86.3%, median age 46	Online, cross sectional, August–September 2020	Professional qualification, concomitant diseases	HLVa Median Total 3.1 Functional 1.8 Inter-critical 3.2	Vaccine confidence index (VCI)	Intention to be vaccinated against flu	**Vaccine confidence completely mediated the effect between ICVL and flu vaccine intention**
Han et al. [30] 2023	Factors Influencing Human Papillomavirus Vaccination Among Asian Immigrant College Students During the COVID-19 Pandemic	USA, college students	133, females 69.9%, mean age 25.12 ± 5.38 years	Cross-sectional from June through August 2021	No association investigated	HPV VL Scale, 3.31 ± 1.83	11-item HPVattitude scale; 4-item HPVvaccine norms scale; 3-item HPV Self-efficacy scale; HPV VH and vaccine intention	HPV Vaccination	*Vaccine subjective norms and literacy directly affected vaccination intention. Vaccine attitudes and self-efficacy directly and negatively affected VH*.
Iskender et al. [31] 2023	The effect of COVID-19 Vaccine Literacy on Attitudes towards COVID-19 Vaccine among University Students	Turkey, students	2384, female 1574, mean age 21.77 years	Cross-sectional survey online September–October 2021	Socioeconomic level parents’ education, COVID-19 diagnosis	COVID-19 VLS FUVL 10.04 ICVL 17.22 (summative score)	None investigated	Attitudes towards COVID-19-VLS (nine items, two subscales (positive attitude and negative attitude)	Low levels of correlation between VL and attitudes towards vaccine
Kerkez et al. [32] 2023	An Assessment on the Knowledge and Attitudes of University Students Concerning Adult Immunization and COVID-19 Vaccine in Turkey	Turkey, students	307 females 52.4%, mean age 20.4 ± 0.56 years	Cross-sectional from June through August 2021	No association investigated	COVID-19 VLS, FUVL 2.40 ± 0.71 ICVL 2.93 ± 0.81	None investigated	Attitudes toward the COVID-19 vaccine scale; Knowledge for adult vaccines	*VL level contributed positively to adult vaccine knowledge level and attitude toward the COVID-19 vaccine*
Lu et al. [33] 2023	Lessons Learned from COVID-19 Vaccination Implementation: How Psychological Antecedents of Vaccinations Mediate the Relationship between Vaccine Literacy and Vaccine Hesitancy	China, general population	1015, female 53.3%,	April 2021	Income, place of residence, marital status	COVID-19 VLS Score range 1–5, Low hesitant FUVL 3.8 ICVL 3.56 High hesitant FUVL 3.6 ICVL 3.24	“3Cs” psycholo= gical antecedents of vaccination	COVID-19 vaccine uptake; 11-point self-reported scale on ‘3Cs’; 10-point VH visual scale on non-vaccinated particiipants	**“3Cs” psychological antecedents were significant mediators between VL (mainly ICVL) and VH;** **Time-to-event analysis confirmed the role of VH in delaying vaccination**
Maneesriwongul et al. [34] 2023	Parental Vaccine Literacy: Attitudes towards the COVID-19 Vaccines and Intention to Vaccinate their Children Aged 5–11 Years against COVID-19 in Thailand	Thailand, parents	542, female 83.2%, 60.9% between ages 36 and 45 years	Online cross-sectional study, from January to February 2022	Income sufficiency, occupation, child’s age, underlying diseases, parents’ vaccinat status	COVID-19 VLS FUVL 2.67 ± 0.69 ICVL 3.31 ± 0.51	Parents’ attitudes towards COVID-19 vaccine (10 questions)	Parents’ intention to have children vaccinated against COVID-19	*Factors influencing intention to vaccinate were: child age,* *parents’ education,* *ICVL,* *positive attitudes* *toward vaccine*
Maneesriwongul et al. [35] 2023	Parental Hesitancy on COVID-19 Vaccination for Children Under Five Years in Thailand: Role of Attitudes and Vaccine Literacy	Thailand, parents	455, female 83.7%, 55.8% <35 years	Online cross-sectional study	Income sufficiency, occupation, child’s age, underlying diseases, parents’ vaccination status	COVID-19 VLS FUVLl 2.8 ± 0.71 ICVL 3.3 ± 0.56	Parents’ attitudes towards COVID-19 vaccine (10 questions)	Parents’ intention to have children vaccinated against COVID-19	*Factors influencing intention to vaccinate were: parents’ age > 35, education, income,* *ICVL,* *positive attitudes* *toward vaccine*
Montagni et al. [36] 2022	Measuring Digital Vaccine Literacy: Development and Psychometric Assessment of the Digital Vaccine Literacy Scale	France, adults	848, females 73.1%, mean age 29.9 ± 12.3 years	Cross sectional validation study	Field of study	Digital vaccine literacy scale Score 19.5 ± 2.8	None investigated	Flu vaccination, source of vaccine-related information	Digital vaccine literacy tool showed good psychometric proprieties
Shon et al. [37] 2023	Effects of Vaccine Literacy, Health Beliefs, and Flu Vaccination on Perceived Physical Health Status among Under/Graduate Students	USA, Students	382, females 73.8%, mean age 22.37 ± 5.97 years	Web-based survey, September 2019 to March 2020	Family income, parents’ education, Insurance, Race	VL: single question on flu vaccine, nominal scale	Health Beliefs (HBM scale, 16 questions)	Flu vaccine uptake (seelf reported)	**Results showed direct effect of VL on flu vaccine uptake, and mediating effects of health beliefs (benefit, severity and susceptibility) between VL and vaccination**
Us et al. [38] 2023	Turkish Parents’ Attitudes towards COVID-19 Vaccination of their Children aged 12–17 Years: A Cross-Sectional Study: Parents’ Attitudes to COVID-19 Vaccination	Turkey, parents	259 female 81.9%, mean age 41.93 ± 5.68 years	Online cross-sectional	No association was investigated	COVID-19-VLS Total 2.61 ± 0.55, FUVL 2.64 ± 0.83 ICVL 2.60 ± 0.71	Perception of Control of Covid Scale, and of Causes of COVID-19, Attitudes vs the COVID-19 Vaccine Scale	Children vaccination status	*VL increasedboth the reduction in misconceptions and the positive effect on families’ vaccination attitudes*
Yang et al. [39] 2023	Assessing Vaccine Literacy and Exploring its Association with Vaccine Hesitancy: A Validation of the Vaccine Literacy Scale in China	China, adults	12,586, females 43.9%, mean age 31.56 ± 9.12 years	Online, cross-sectional validation study, May 2022 to June 2022	No association investigated	HLVa range 1–5 functional 3.23 ± 1.24, interactive 4.03 ± 0.81, critical 4.03 ± 0.84	VH	Vaccine acceptance	*People who scored lower on the functional scale were more likely to be hesitant in all vaccine acceptance subgroups*
Yilmazel et al. [40] 2023	Attitudes towards COVID-19 Vaccination, Vaccine Hesitancy and Vaccine Literacy among Unvaccinated Young Adults	Turkey, adults	860, females 67.7%, mean age 22.9 ± 3.3 years	Cross-sectional January to April 2021	No association investigated	COVID-19-VLS 27.3 ± 6.5 (summative score)	None investigatd	Vaccine hesitancy scale in pandemics, Attitudes towards COVID-19 vaccine	*Pandemic vaccine hesitancy coincided with low VL and negative attitudes towards vaccines *
Yorulmaz et al. [41] 2023	A Vaccine Literacy Scale for Childhood Vaccines: Turkish Validity and Reliability Vaccine Literacy Scale	Turkey, parents	285, females % not reported, mean age 34.7 ± 6.6 years	Online, cross-sectional validation study, From 25 May 2022 to 25 June 2022	No association investigated	Vaccine Literacy Scale (Aharon et al., 2017 [42])	None investigated	Health Literacy Scale (HLS-14)	There was a negative correlation between the Vaccine Literacy Scale and HLS-14

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
