# Peer review of "Perspectives in the Development of Tools to Assess Vaccine Literacy"

_vaccines, 2024, doi:10.3390/vaccines12040422_

Round 1

Reviewer 1 Report

Comments and Suggestions for Authors

Interesting but how difficult to read: some sentences exceed 8 or 10 lines, many initialisms and sometimes without any explanation in the text.

Just for smiling line 697: Don't forget that females are fully human and that parents want to protect them by vaccination! 

Author Response

We thank the reviewer for the comments. Below our responses to the respective comments:

Reviewer’s comment: Interesting but how difficult to read: some sentences exceed 8 or 10 lines, many initialisms and sometimes without any explanation in the text.

Our response: we have simplified the wording, trying to make the English text more fluid. As for acronyms, we kept a minimum number.

Reviewer’s comment: Just for smiling line 697: Don't forget that females are fully human and that parents want to protect them by vaccination! 

Our response: We imagine it refers to the Phrase: “The controversial nature of coronavirus vaccination has exacerbated the pressure on parents who make decisions about their sons’ immunization.” We have corrected it into “sons and daughters’ ”. Sorry for the inaccuracy…!!

Note: According to other Reviewers’ comments, and to facilitate Vaccines’ reader, we have redefined objectives and Methods. Title ad abstract have been modified accordingly.

Objectives now are: "The objective of this study is to suggest a framework including the different visible or latent factors underlying VL, and related variables, to be used as a basis of the development and validation of future assessment tools. In pursuing this objective, an updated review of existing research on VL and its measures has been performed, in addition to a post-hoc analysis of a previous dataset."

Reviewer 2 Report

Comments and Suggestions for Authors

In the title the authors announced a synthesis of the literature on vaccine literacy (VL). In the abstract they stated to follow the publication of recent scoping and systematic reviews of existing literature on vaccine literacy (VL), its relationship to vaccine hesitancy (VH), and associated variables, such as vaccination beliefs, attitudes, and behaviors towards immunization. The objective seemed to be to present a theoretical framework, where the positioning of VL at the intersection between antecedents and beliefs/attitudes leading to behaviors explains why and how VL serves as a tool and a critical, direct, or mediating driver of vaccination choices, overcoming VH and increasing vaccination rates, operating at personal, as well as at organizational and community level. However, they presented two objectives in the text’s body: the first, to provide an overview of existing research on VL and its measures; and the second, to deepen the understanding of the different visible or latent factors underlying VL and to suggest a framework as a basis for the development and validation of future tools. They sought to undertake an overview but the adopted procedures are not clear. As the authors stated that there are four reviews (a meta-analysis, two systematic reviews and a scoping review), I am afraid that the justification of the study would need to be consistently presented. According to the Methods, the authors updated previous review focusing at tools and statistical methods used. After, they performed a post-hoc evaluation of data from a mid-2020 survey through mediation and factor analysis, where variables were relabeled to improve the understanding of the relationships between them. Next they elaborated a theoretical framework based on the literature review, and on a backward citation search. I am afraid that the manuscript’s format brings difficulties to the Vaccines’ reader. It is difficult to understand why only studies published in 2023 were included if the objective was to update the literature synthesis. It is not clear the criteria used for selecting the datasets submitted to statistical analysis. It is relevant to propose theoretical framework but the method needs to be fully reformulated.

Comments on the Quality of English Language

no comments

Author Response

We thank the Reviewer for the comments. Below our responses to the respective comments:

Reviewer’s comment:  In the title the authors announced a synthesis of the literature on vaccine literacy (VL). In the abstract they stated to follow the publication of recent scoping and systematic reviews of existing literature on vaccine literacy (VL), its relationship to vaccine hesitancy (VH), and associated variables, such as vaccination beliefs, attitudes, and behaviors towards immunization. The objective seemed to be to present a theoretical framework, where the positioning of VL at the intersection between antecedents and beliefs/attitudes leading to behaviors explains why and how VL serves as a tool and a critical, direct, or mediating driver of vaccination choices, overcoming VH and increasing vaccination rates, operating at personal, as well as at organizational and community level. However, they presented two objectives in the text’s body: the first, to provide an overview of existing research on VL and its measures; and the second, to deepen the understanding of the different visible or latent factors underlying VL and to suggest a framework as a basis for the development and validation of future tools. They sought to undertake an overview, but the adopted procedures are not clear. As the authors stated that there are four reviews (a meta-analysis, two systematic reviews and a scoping review), I am afraid that the justification of the study would need to be consistently presented. According to the Methods, the authors updated previous review focusing at tools and statistical Methods used. After, they performed a post-hoc evaluation of data from a mid-2020 survey through mediation and factor analysis, where variables were relabeled to improve the understanding of the relationships between them. Next they elaborated a theoretical framework based on the literature review, and on a backward citation search. I am afraid that the manuscript’s format brings difficulties to the Vaccines’ reader.

Our response: in order to facilitate Vaccines’ reader, we have redefined objectives and Methods. The proposal of a theoretical framework is now taken as the objective of the study, with the subsequent development of new possible VL instruments. The search for articles published in 2023 is considered an update of the relevant literature on the subject, as our earlier scoping review was until 2022 (Biasio et al. Hum Vaccin Immunother. 2023;19(1):2176083). Completing the review of the literature was needed before proposing the framework, therefore, it has been kept in Methods. We also believe that it might be better to combine the literature review, the framework proposal, followed by the development of future VL tools into one publication (instead of separate papers). This would provide a comprehensive exploration of the topic for the reader of Vaccines. We recognize that the text can be difficult to follow at times, so we have made efforts to revise, reduce and improve it.

Reviewer’s comment: It is difficult to understand why only studies published in 2023 were included if the objective was to update the literature synthesis.

Our response: For the purposes of our research, we did not consider only studies published in 2023, but all previous literature on the topic. As mentioned, the 2023 literature search was meant to complement the work that had already been published until the end of 2022 in the previous scoping review (see above). Considering the construction of the framework as the main objective of the study, the literature search of the 2023 publications should not appear as the main objective. Therefore, we have reformulated the objectives as follows:

"The objective of this study is to suggest a framework including the different visible or latent factors underlying VL, and related variables, to be used as a basis of the development and validation of future assessment tools. In pursuing this objective, an updated review of existing research on VL and its measures has been performed, in addition to a post-hoc analysis of an earlier dataset."

Consequently, the title of the manuscript has been simplified and changed into:

"Perspectives in the Development of Tools to Assess Vaccine Literacy"

Reviewer’s comment:  It is not clear the criteria used for selecting the datasets submitted to statistical analysis. It is relevant to propose theoretical framework but the method needs to be fully reformulated.

Our response: The dataset was selected because it refers to a study (Biasio et al Hum Vaccin Immunother. 2021;17(5):1304-12) where a VL tool was used (Covid-19-VLS) which can been taken as an example for the development of new instruments. We have chosen this tool also because it has been adapted and used by several investigators and translated in different languages (as reported in the manuscript), while keeping the same construct in all its versions and languages. Therefore, it is known by different investigators in the field, and, we think, by those Vaccines‘ readers interested in Health and Vaccine Literacy.

The wording about the dataset has been modified into: "In addition, we carried out a post-hoc analysis from the dataset of our survey conducted in 2020 in which a sample of 885 individuals of the general population was recruited to fill out an online questionnaire to evaluate VL levels on Covid-19. The data series from this study was chosen for the analysis because the same assessment scale has been translated into different languages and used in various populations" (lines 126-128 of the revised manuscript).

Methods have been reformulated, following Reviewer’s suggestion.

The revised manuscript - with the cover letter - is attached

Reviewer 3 Report

Comments and Suggestions for Authors

The authors aimed to provide an overview of existing research on VL and its measures, as a kind of state of the art of this specific topic of preventive medicine. A second objective, based on selected literature was to deepen the understanding of the different visible or latent factors underlying VL and to suggest a framework as a basis for the development and validation of future tools.

The study covers some issues that have been overlooked in other similar topics. The structure of the manuscript appears adequate and well divided in the sections. Moreover, the study is easy to follow, but few issues should be improved. Some of the comments that would improve the overall quality of the study are:

a. Authors must pay attention to the technical terms acronyms they used in the text.

b. Conclusion Section: This paragraph required a general revision to eliminate redundant sentences and to add some "take-home message".

Author Response

We thank the Reviewer for the comments. Below our responses to the respective comments:

Reviewer’s comment:  The authors aimed to provide an overview of existing research on VL and its measures, as a kind of state of the art of this specific topic of preventive medicine. A second objective, based on selected literature was to deepen the understanding of the different visible or latent factors underlying VL and to suggest a framework as a basis for the development and validation of future tools.

Our note: according to the comments of other Reviewers, we have reformulated the objectives, to provide more relevance to the framework, as follows:

"The objective of this study is to suggest a framework including the different visible or latent factors underlying VL, and related variables, to be used as a basis of the development and validation of future assessment tools. In pursuing this objective, an updated overview of existing research on VL and its measures has been performed, in addition to a post-hoc analysis of an earlier dataset.."

Reviewer’s comment:  The study covers some issues that have been overlooked in other similar topics. The structure of the manuscript appears adequate and well divided in the sections. Moreover, the study is easy to follow, but few issues should be improved. Some of the comments that would improve the overall quality of the study are:

  1. Authors must pay attention to the technical terms acronyms they used in the text.
  2. Conclusion Section: This paragraph required a general revision to eliminate redundant sentences and to add some "take-home message"

Our response: we have revised Conclusions, according to the Reviewer's comments. We reduced acronyms, keeping a minimum number.

Reviewer 4 Report

Comments and Suggestions for Authors

Uptake of vaccines is an important public health goal. Understanding factors that influence vaccine uptake is important. This manuscript reports a lot of stuff related to this, but includes so many things that the value is lost. A few critical issues to consider:

The title is misleading - there are reviews, post-hoc survey data analysis, scale development, model development, opinions...  I'm not sure what this paper is but it is not a scoping review.

Intro lines 55-57: "... varying levels of HL may lack the necessary skills pertaining to vaccination" I think you mean 'higher' levels of HL

Intro lines 57-58: Citation for this??

Intro - citation 10 Zhang et al did abstract and report on the measurement tools for VL... so why is the current paper needed?

Citations 19 and 20 are the same, right? 

The authors justify the current project by saying "Various instruments have been developed to assess VL, although they  may be considered limited by the number and complexity ..." but then say "there is a need to develop new, more comprehensive tools..."  If you are going to create something that is more comprehensive, how is it less complex?? 

Line 72 - "VL levels in the population..." What population? Are the authors assuming that VL is homogenous across the globe? Vaccine uptake challenges are different in different populations/communities. 

Methods - These are very confusing. Is this a scoping review, a systematic review, some other kind of review? The results indicate it was more like a narrative review, in which the authors 'selected' articles to include. That is not how a scoping review (nor a systematic review, nor a systematic scoping review) works. The citations are outdated and the current guidance was not utilized. The citation to the PRISMA guideline is incorrect. It should be to Tricco et al 2018. Why is the original scoping review methodology paper by Arksey and O'Malley even cited? Where is the protocol for the review? The purpose of this study indicates a scoping review is not an appropriate methodology. It appears this is meant to be an update of a systematic review. In that case it would be a systematic review. However, the authors also indicate that they may not have followed the methods for a systematic review... so I'm not sure what this is. 

2.2 Statistical and post-hoc analysis - lines 122-124 - this is not appropriate for a scoping review

Lines 125-136: What is this? This is not hinted at in the intro, is not in the abstract, is not reflected in the title .... ??? 

Line 146: intention to vaccinated was taken as an attitude... this is not a correct interpretation of what an attitude is, even if it was before the vaccine was available. This is contradictory to the entire health behavior theory literature. 

2.3 - theoretical framework - this is a whole new component and should be a separate paper.

Results - line 160-161 - "VL was assessed using specific tools as described in Methods"  It's not described in the Methods section.

PRISMA figure - this is missing the reasons for records excluded at the screening step of full text review. This is a requirement per PRISMA.

Table 1 - 'selected' studies ... this is not how a scoping nor systematic review work

Results - lines 170-172 - 9 out of 17 'selected for the present review' confirmed the association between VL and vaccine acceptance/intention. That's less than half. That does not support the point of this study that VL is a critical driver of vaccination behavior. 

The results section has so much going on it is hard to follow.

Post-hoc analysis: line 260-261 - multi-mediation model is not described in the methods. The citation is to a prior review not a multi-mediation model method... 

Comments on the Quality of English Language

Some phrasing is difficult to follow, only minor editing is needed. 

Author Response

We thank the Reviewer for the comments. Below our responses to the respective comments:

Reviewer’s comment:  Uptake of vaccines is an important public health goal. Understanding factors that influence vaccine uptake is important. This manuscript reports a lot of stuff related to this,but includes so many things that the value is lost.

Our response: We acknowledge that the manuscript includes a significant amount of information, but we believe that all the arguments presented are relevant, considering that it is intended for the special issue 'Vaccine Literacy and Social-Cognitive Determinants of Vaccination'. In our opinion, it would be more advantageous to combine the literature review, framework proposal, and the development of future VL tools in a single publication rather than separate documents. This would provide a comprehensive exploration of the topic for the reader of Vaccines. It seems to us a bit excessive to assume that value has been lost. We recognize that the manuscript needed revision and simplification. We did, while carefully considering and respecting the comments of all Reviewers, in making changes.

Reviewer’s comment:  The title is misleading - there are reviews, post-hoc survey data analysis, scale development, model development, opinions...  I'm not sure what this paper is but it is not a scoping review.

Our response: We have simplified the title into: "Perspective in the Development of Tools to Assess Vaccine Literacy".

We have also changed the objectives into: 

"The objective of this study is to suggest a framework including the different visible or latent factors underlying VL, and related variables, to be used as a basis of the development and validation of future assessment tools. In pursuing this objective, an updated review of existing research on VL and its measures has been performed, in addition to a post-hoc analysis of an earlier dataset."

A complete review of the literature was necessary before proposing the framework. We had named it “scoping” because it was the prosecution of our previous scoping review, following a similar methodology.  Now it is defined as a literature update.

Abstract also has been revised.

Reviewer’s comment:  Intro lines 55-57: "... varying levels of HL may lack the necessary skills pertaining to vaccination" I think you mean 'higher' levels of HL

Our response: We agree, it was a mistake, we have changed the text accordingly.

Reviewer’s comment:   Intro lines 57-58: Citation for this??

Our response: there is no citation, the definitions themselves (Vaccine Literacy and Vaccine Hesitancy) are explicit. However, in the Discussion, it is reported (and quoted) that the association strength of vaccine confidence with literacy has been shown to be lower when assessed by general HL functional tools, respect to specific vaccine literacy tools.

Reviewer’s comment:   Intro - citation 10 Zhang et al did abstract and report on the measurement tools for VL... so why is the current paper needed?

Our response: we do not completely understand this observation. We believe that contributions from different Authors are important. Even though we share many of Zhang et al’s views, we like to express ours and bring practical proposals for the development of future VL tools. Our work is substantially different and consists in investigating the link between vaccine literacy, determinants, and outcomes, with the aim of building a framework.

Reviewer’s comment:  Citations 19 and 20 are the same, right? 

Our response: Correct: this is due to a malfunction of the references' software, we have tried to fix it (now the correct reference is #16).

Reviewer’s comment:   The authors justify the current project by saying "Various instruments have been developed to assess VL, although they  may be considered limited by the number and complexity ..." but then say "there is a need to develop new, more comprehensive tools..."  If you are going to create something that is more comprehensive, how is it less complex?? 

Our response: probably we have not been correct in using adjectives such as "comprehensive" and "complex". However, we believe that while a comprehensive tool may sometimes be complex, this is not a prerequisite. A tool can be comprehensive in the sense that it covers a broad range of aspects or provides extensive information, without being overly complicated or difficult to use.

However, to avoid misunderstanding, we have changed the second sentence into: “Therefore, there is a need to develop new tools for a more extensive assessment of VL.”  (lines 74-75)

Reviewer’s comment:   Line 72 - "VL levels in the population..." What population? Are the authors assuming that VL is homogenous across the globe? Vaccine uptake challenges are different in different populations/communities. 

Our response:  it is intended that the phrase refers to the respective populations studied. It is obvious that VL differences between populations may exist. However, in the mentioned, previous scoping review, comparisons between populations were performed, using the same assessment scale (Covid-19-VLS), showing comparable findings not only in terms of VL score, but also results of PCA, with statistically similar distribution of the tools’ items load.

Reviewer’s comment:   Methods - These are very confusing. Is this a scoping review, a systematic review, some other kind of review? The results indicate it was more like a narrative review, in which the authors 'selected' articles to include. That is not how a scoping review (nor a systematic review, nor a systematic scoping review) works. The citations are outdated and the current guidance was not utilized. The citation to the PRISMA guideline is incorrect. It should be to Tricco et al 2018. Why is the original scoping review methodology paper by Arksey and O'Malley even cited? Where is the protocol for the review? The purpose of this study indicates a scoping review is not an appropriate methodology. It appears this is meant to be an update of a systematic review. In that case it would be a systematic review. However, the authors also indicate that they may not have followed the Methods for a systematic review... so I'm not sure what this is. 

Our response: as mentioned above, this work contains an update of a previous scoping review.

We would like to address a potential translation error: instead of stating that we 'selected' studies, the accurate term is 'included' studies. The discrepancy may have arisen due to the fact that in Italian language, the respective words for 'selection' and 'inclusion' can sometimes be used interchangeably to describe the process of literature review. We apologize for any confusion this may have caused. The necessary corrections have been made within the text.

We have revised our references per Reviewer’s suggestions and replaced the citation of the PRISMA guideline with that of Tricco et al. (2018), as recommended.

Regarding the protocol for the review, we would like to emphasize that while a study protocol is a recommended practice, it is not mandatory for a scoping review.

Reviewer’s comment:   2.2 Statistical and post-hoc analysis - lines 122-124 - this is not appropriate for a scoping review

Our response: we believe this observation is overcome by the fact that we have redefined the objectives and the methodology: a literature review update is now taken as a prerequisite of the framework creation, together with the post-hoc analysis. However, describing the demographics and VL score of the articles included in the review seems to us correct.

Reviewer’s comment:   Lines 125-136: What is this? This is not hinted at in the intro, is not in the abstract, is not reflected in the title .... ??? 

Our response: the post-hoc analysis is now mentioned also in Introduction and in the abstract.

Reviewer’s comment:   Line 146: intention to vaccinated was taken as an attitude... this is not a correct interpretation of what an attitude is, even if it was before the vaccine was available. This is contradictory to the entire health behavior theory literature. 

Our response: we mean that intentions are precursor of behaviors, and are directly influenced by attitudes. Limbu et al (Vaccines 2022;10(12):2026) reviewed the application of Theory of Planned Behavior in predicting vaccination intention against Covid-19, showing that attitudes had the strongest association with vaccination intention, followed by subjective norms, and perceived behavioral control. The same has been observed by Wolff (Front Psychol. 2021;12:648289),

Our assumption was also based on the APA definitions of attitude (“relatively enduring and general evaluation of an object, person, group, issue, or concept on a dimension ranging from negative to positive”) and behavior (“any action or function that can be objectively observed in response to controlled stimuli”: in our case the action was = getting vaccinated against Covid-19). When our survey was conducted (mid 2020) there was no direct opportunity for individuals to engage in the behavior of getting vaccinated. The availability of vaccines came several months later.

In addition, PCA conducted on the 2020 dataset showed that the item related to Covid-19 vaccine intention load on the same factor than beliefs about vaccination (number of components based on parallel analysis), supporting the observation that it was closer to attitudes than behaviors, while behavior toward flu vaccination (vaccine uptake) loaded on a different factor. Data and results are available on request.

However, we have simplified the sentence into: “For the purposes of the post-hoc analysis, last seasonal flu vaccine received (self-reported) was considered as behavior/outcome..” (lines 147-148 of the current version of the manuscript).

In fact, for the purpose of our statistical 'exercise', in particular in exploring the mediating role of variables between VL and outcomes, we needed to use one consolidated behavior, and flu uptake was considered the most reliable.

Reviewer’s comment:   2.3 - theoretical framework - this is a whole new component and should be a separate paper.

Our response: as mentioned, in revising the manuscript, we have given relevance to the framework, and modified the objectives’ statement. We also felt that by including in the same publication the literature update, the proposal of a framework and the development of future VL tools could allow a complete exploration on this specific topic, for the convenience of the Vaccines’ reader.

Reviewer’s comment:  Results - line 160-161 - "VL was assessed using specific tools as described in Methods"  It's not described in the Methods section.

Our response: the sentence has been changed into: “VL was assessed using specific tools.”

Reviewer’s comment:  PRISMA figure - this is missing the reasons for records excluded at the screening step of full text review. This is a requirement per PRISMA.

Our response: the Prisma figure has been changed according to Reviewer’s suggestion.

Reviewer’s comment:  Table 1 - 'selected' studies ... this is not how a scoping nor systematic review work.

Our response: we believe that charting the ‘included’ studies is correct for reviews.

Reviewer’s comment:  Results - lines 170-172 - 9 out of 17 'selected for the present review' confirmed the association between VL and vaccine acceptance/intention. That's less than half. That does not support the point of this study that VL is a critical driver of vaccination behavior. 

Our response: Nine publications out of 17 confirmed a direct association between VL and vaccine acceptance/intention, while other three papers confirmed a completely or partially mediated association (as reported later in the manuscript), and only two reported opposite findings. Furthermore, the results are like those of two systematic reviews, as underlined in the text.

We adapted the text for a better understanding. (lines 342-344 of the revised manuscript).

However, the fact that not all publications included the review confirmed an association between VL and VH, does not necessarily mean that VL cannot be considered a critical driver of vaccination behavior. It may depend on the objectives, methodology and endpoints of the single studies, in addition to how the statement “driver of vaccination behavior” is understood, i.e. whether VL is considered a direct or mediating variable, or both.

The recent meta-analysis by Isonne et al (Hum Vaccin Immunother. 2024;20(1):2321675.) confirmed VL as a strong predictor of vaccination intention, while its association with vaccination status (vaccine uptake) was weaker. This matches with the positioning we have attributed to VL in the framework, at the intersection between antecedents and attitudes, leading to behaviors and explaining why VL serves as a tool and a critical, direct, or mediating driver of vaccination choices. Due to these reasons we define VL as a “…driver towards vaccine intention, …” (line 415 of the revised manuscript).

Moreover, Zhang et al (quoted above by the Reviewer) define VL as “an important ‘endogenous driver’ of people’s vaccine choices, overcoming vaccine hesitancy and increasing vaccination rates.”

Reviewer’s comment:  The results section has so much going on it is hard to follow.

Our response: we apologize, but we do not understand the Reviewer’s recommendation. If this is related to too much text or complex sentences, we have revised, reduced, and simplified the text.

Reviewer’s comment:   Post-hoc analysis: line 260-261 - multi-mediation model is not described in the Methods. The citation is to a prior review not a multi-mediation model method..

Our response: multi-mediation model description has been added in Methods. Citation corresponds to that reported at the software web page.